# Pathogen at the Gates: Human Cytomegalovirus Entry and Cell Tropism

**DOI:** 10.3390/v10120704

**Published:** 2018-12-11

**Authors:** Christopher C. Nguyen, Jeremy P. Kamil

**Affiliations:** Department of Microbiology and Immunology, Louisiana State University Health Sciences Center, Shreveport, LA 71103, USA; cnguy7@lsuhsc.edu

**Keywords:** viral entry, glycoproteins, receptors, cytomegalovirus, herpesviruses, HCMV, CMV, pentamer, trimer, gH/gL

## Abstract

The past few years have brought substantial progress toward understanding how human cytomegalovirus (HCMV) enters the remarkably wide spectrum of cell types and tissues that it infects. Neuropilin-2 and platelet-derived growth factor receptor alpha (PDGFRα) were identified as receptors, respectively, for the trimeric and pentameric glycoprotein H/glycoprotein L (gH/gL) complexes that in large part govern HCMV cell tropism, while CD90 and CD147 were also found to play roles during entry. X-ray crystal structures for the proximal viral fusogen, glycoprotein B (gB), and for the pentameric gH/gL complex (pentamer) have been solved. A novel virion gH complex consisting of gH bound to UL116 instead of gL was described, and findings supporting the existence of a stable complex between gH/gL and gB were reported. Additional work indicates that the pentamer promotes a mode of cell-associated spread that resists antibody neutralization, as opposed to the trimeric gH/gL complex (trimer), which appears to be broadly required for the infectivity of cell-free virions. Finally, viral factors such as UL148 and US16 were identified that can influence the incorporation of the alternative gH/gL complexes into virions. We will review these advances and their implications for understanding HCMV entry and cell tropism.

## 1. Introduction

HCMV exhibits a broad cell tropism that is reflected in the multifarious tissues and organs in which the virus causes disease [1]. Despite evidence that the virus alters its gH/gL complexes in a manner that depends on the cell-type producing virus [2], the regulation of HCMV cell tropism is poorly understood. This review will focus on HCMV cell tropism as it relates to the viral entry machinery found in the virion envelope. Over the last several years, substantial progress has been made in this area. A number of new cellular receptors have been identified, and in two key examples, the newly identified cellular receptor is matched to a specific gH/gL complex. Furthermore, a new gH complex has been characterized, and a number of viral factors that contribute to strain-specific differences in cell tropism have been described. Although there remain important unresolved questions, these advances provide a new level of clarity for understanding HCMV cell tropism and suggest new models to explain how HCMV enters cells and spreads within tissues.

## 2. Viral Envelope Glycoprotein Complexes and Receptors

Before the first complete HCMV genome sequence was published in 1990 [3], three major disulfide-linked viral envelope glycoprotein complexes had been described [4,5,6]. These complexes, originally designated as gC-I, gC-II, and gC-III, have turned out to play crucial roles in HCMV entry. The viral genes encoding the constituents of each complex are now known, as are the relationships to the conserved entry machinery shared among all herpesviruses. Therefore, the complexes are increasingly referred to by terms shared across the *Herpesviridae*, e.g., glycoprotein H/glycoprotein L (gH/gL), glycoprotein B (gB), glycoprotein M/glycoprotein N (gM/gN). When a constituent gene product is unique to the betaherpesviruses or to HCMV, the name of that product is often used, e.g., gO (UL74) in the case of gH/gL/gO for trimer, or UL116 in the case of gH/UL116, the newly discovered gH complex with UL116 [7].

## 3. gB

gC-I is made up of homotrimers of glycoprotein B (gB), a pan-*Herpesviridae* conserved glycoprotein posited to serve as the proximal mediator of membrane fusion events during viral entry. The three-dimensional structures of post-fusion gB from herpes simplex virus-1 (HSV-1), HCMV, and the Epstein–Barr virus resemble those of glycoprotein G from the rhabdovirus vesicular stomatitis virus (VSV G) and of gp64 from the *Autographa californica* nuclear polyhedrosis virus, a baculovirus [8,9]. Together, VSV G, gp64, and gB comprise the class III membrane fusogens [10]. Based on inferences from the pre-fusion structure of VSV G, gB is thought to dramatically rearrange during membrane fusion. In its pre-fusion form, gB is thought to adopt a relatively flattened conformer in which the fusion loops are positioned at the base of the homotrimer, close to the viral membrane—hence, tucked away from the target membrane and set apart from one another. In the prevailing model, fusion occurs via a transitory intermediate in which the fusion loops reach out to the target membrane [10]. In the post-fusion configuration, three central helices line up at the core of the homotrimer, elongating the structure, and causing the fusion loops to cluster closer together at the side of the homotrimer opposite from where they began [11].

HCMV gB, which is encoded by *UL55*, is synthesized as a 160-kD precursor that undergoes furin cleavage in the Golgi, resulting in 116 kD and 55 kD fragments that remain disulfide-linked to each other [12]. In 2015, two crystal structures of the post-fusion form of the HCMV gB ectodomain were published, both at up to 3.6 Å resolution [9,13]. In one of the structures, the ectodomain was crystallized in complex with a recombinant Fab from a neutralizing antibody [9]. In a noteworthy development, a very recent cryo-electron tomography (cryoET) study reports the visualization at up to 21 Å resolution of pre-fusion and post-fusion gB conformers on the envelope of native HCMV virions [14]. The pre-fusion gB homotrimer was described as a squat “Christmas tree-shaped” density (extending 130 Å out from the membrane) compared to post-fusion homotrimers, which appeared taller and more columnar (extending 161 Å from the membrane).

A number of cell-surface proteins have been reported or implicated as gB receptors, including the epidermal growth factor receptor (EGFR) [15], the platelet-derived growth factor receptor alpha (PDGFRα) [16], and, as discussed further below, integrins [17,18]. On the other hand, it has also been suggested that gB functions as a viral fusogen that does not bind cellular receptors [19]. In light of the latter, attempts would seem warranted to visualize interactions between gB and its putative receptors by approaches such as cryo-electron microscopy (cryo-EM).

## 4. gM/gN

gC-II is comprised of a disulfide-linked heterodimer of glycoproteins M (gM) and N (gN), which are encoded by *UL100* and *UL73*, respectively [20,21]. gM/gN is the most abundant glycoprotein complex on virions [22,23] and is essential in HCMV, as null mutants are non-viable. The gM/gN complex plays key roles during attachment to host cells, likely by mediating interactions with heparan sulfate proteoglycans on the cell surface [24]. Notably, gM/gN also plays intracellular roles during viral replication that are independent of its roles in attachment [25,26]. gM is an *N*-glycosylated 48-kD type III transmembrane (TM) glycoprotein with seven predicted TM helices, while gN is a single-pass type I TM protein that is extensively *O*-glycosylated. The unmodified 138 amino acid gN polypeptide from strain AD169 specifies a molecular weight of approximately 18 kD when expressed on its own; however, the fully glycosylated (mature) form detected from virion lysates migrates at ~65-kD in sodium dodecyl sulfate polyacrylamide gel electrophoresis (SDS-PAGE) [20,21]. Motifs in the gM cytoplasmic tail are required for trafficking during virion assembly [25], and the cytoplasmic tail of gN, which is palmitoylated at two different cysteine residues, is required for secondary envelopment [26]. The gN coding sequence varies remarkably across HCMV strains [27,28], consistent with the observation that gM/gN is an important target for humoral immune responses [29].

## 5. The Trimeric gH/gL Complex and its Receptors

gC-III, now frequently referred to as the “trimer” or “gH/gL/gO,” is a heterotrimeric complex, in which the heterodimer of gH (UL75) and gL (UL115) is disulfide-linked to glycoprotein O (gO), a heavily *N*-glycosylated polypeptide encoded by *UL74* [30,31,32]. All herpesviruses encode gH/gL complexes, as gH/gL and gB together comprise the “core” herpesvirus membrane fusion machinery. Homologs of gO, in contrast, are found only among betaherpesviruses. The emerging consensus is that gO, in the context of trimer, is required for the infectivity of cell-free virions [33,34,35]. The platelet-derived growth factor receptor alpha (PDGFRα) was identified in three independent studies to function as a cellular receptor for trimer [36,37,38] (Figure 1, Table 1). This finding has continued to find support in the literature [39,40], and the latest data suggest that tyrosine kinase activity of PDGFRα is dispensable for its role in HCMV entry [37,39].

The severe entry defects observed during infection of fibroblasts lacking PDGFRα phenocopy those seen with *gO*-null mutant viruses, with the residual low-level infectivity being pentamer-dependent [33,35,37,38,39]. Thus, a role for PDGFRα in trimer-dependent entry may explain why gO is required for cell-free virions to infect fibroblasts, which ordinarily express PDGFRα. Why gO is required for cell-free virions to infect epithelial or endothelial cells, however, remains unclear, since these cell types either do not express PDGFRα [37], or express it at only low levels that are not necessary for soluble recombinant trimer to bind to the cell surface [41]. Additional hitherto unidentified cellular receptors, or receptor-independent roles in membrane fusion, may explain why the trimer is required for cell-free virus to infect cells that lack PDGFRα.

## 6. The Pentameric gH/gL Complex and its Receptors

In 2005, a second HCMV gH/gL complex, now often referred to as the “pentamer,” was discovered after the repair of a frame-shift mutation in *UL131* (*UL131A*) dramatically expanded the cell tropism of the fibroblast-adapted laboratory HCMV strain AD169, restoring its infectivity for epithelial and endothelial cells [42,43]. The pentamer is composed of the gH/gL heterodimer bound to a trio of small glycoproteins encoded by *UL128*, *UL130*, and *UL131* (also known as *UL131A*) [30,43,44,45]. The *UL128–UL131* locus was observed to be: (i) unstable during HCMV passage in fibroblasts [46], and (ii) required for infection of leukocytes, dendritic cells, epithelial cells, and endothelial cells [47,48,49,50]. The latter observations may have hastened the discovery of the pentamer.

In 2015, a group from GSK Vaccines further defined the assembly of the pentamer. These investigators identified that the cysteine at amino acid position 144 (Cys144) of the gL polypeptide chain forms a disulfide bond to either UL128-Cys162 or gO-Cys351 [30]. These findings explain why the two gH/gL complexes are mutually exclusive. The same study also provided low-resolution EM images of recombinant pentamer and trimer bound to gH antibodies. A subsequent study characterized neutralizing antibody binding sites using similar approaches [51].

In 2017, X-ray crystal structures for the pentamer bound to two different neutralizing antibodies were reported at 3.0 Å and 5.9 Å [45]. Several aspects of the gH domain structure closely resemble Epstein–Barr virus (EBV) gH, while the overall structure is nonetheless described as an intermediate between the “rod-like” conformation of herpes simplex virus-2 gH/gL, and the “boot-like” conformation of EBV gH/gL. Two disulfide bonds connect the N-termini of gH and gL to each other: gH-Cys59 to gL-Cys54, and gH-Cys95 to gL-Cys47. As predicted from the literature [46,47,52,53,54,55], UL128, UL130, and a C-terminal region of gL adopt chemokine folds: of the CC-type for gL and UL128, and of the C-type for UL130, which may suggest that the ancestral cytomegalovirus “pirated” host chemokine genes on multiple occasions. Integration of chemokines into the viral cell entry machinery may have provided receptor binding and signaling properties of immediate benefit to the virus, even if many of these features were later lost or extensively modified during evolution. Another striking aspect of the pentamer structure is how UL128 connects to gL. A ~40 amino acid region of UL128 (residues 123 to 162) forms a surprisingly long (~50 Å) flexible linker that stretches across UL130 and UL131 to reach gL, at which point the chain makes three alpha helical turns and presents Cys162 to form its disulfide linkage to Cys144 of gL [45]. It is fascinating to consider in what order the subunits of the pentamer must assemble for UL128 to adopt this peculiar final conformation.

Earlier this year, neuropilin-2 (Nrp2) was convincingly identified as a functional cell entry receptor for the pentamer [40] (Figure 1, Table 1). To identify Nrp2, the investigators made use of a high-throughput “avidity-based extracellular interaction screen” (AVEXIS), in which recombinant single-pass transmembrane proteins were monitored in vitro for interactions with recombinant trimer and pentamer. After identifying a high-affinity interaction between Nrp2 and the pentamer, the investigators demonstrated that Nrp2 is essential for pentamer-dependent HCMV infection of endothelial and epithelial cells. The screen also identified interactions with other cellular molecules that may represent additional receptors. For trimer, the additional hits included transforming growth factor beta receptor type 3 (TGFβRIII) and neuregulin-2 (NRG2). For pentamer, the additional high-affinity interaction hits were thrombomodulin (THBD), leukocyte immunoglobulin-like receptor subfamily B member 3 (LILRB3), and the immunoglobulin alpha Fc receptor (FCAR). Another hit for the pentamer, though of lower affinity, was CD46. Although the biological relevance for these other hits remains to be established, it seems likely that at least some of these molecules will turn out to play roles during natural infection.

## 7. Additional Receptors

CD147 was recently shown to be required for pentamer-dependent entry into epithelial cells in a manner that does not involve a direct interaction with the pentamer [56] (Figure 1, Table 1). Lujo virus, an arenavirus, requires Nrp2 as a surface receptor, and also requires CD63, a tetraspannin protein, as an intracellular factor for entry [57]. Notably, another tetraspannin, CD151, was also recently reported to play roles during HCMV entry [58]. By analogy, it seems plausible that Nrp2 functions as the proximal cell surface receptor for the pentamer, while other molecules, such as CD147 or CD151, function later as co-receptors, perhaps at a post-internalization step.

Another cellular molecule recently implicated as an HCMV receptor is THY-1 (CD90), which reportedly interacts with both gH and gB [59,60]. THY-1 engages αVβ3 integrins and recruits the signaling adaptor molecule paxillin during signaling. The αVβ3 integrins reportedly function as gH-dependent co-receptors [61], and paxillin has been found to be important during entry into monocytes [62]. Notably, integrins α2, α6, and β1 are also reported to play roles during HCMV entry at a post-attachment step [17,18]. The interactions with integrins are thought to involve a “disintegrin-like” gB motif that resembles motifs found in the integrin-binding domain of cellular proteins of the “a disintegrin and a metalloproteinase” (ADAM) family. Nonetheless, the disintegrin-like motif is mostly buried in the post-fusion gB structure [13]. A high-resolution pre-fusion gB structure might help to shed further light on the role of this motif in entry.

## 8. Many Are Called, Few Are Chosen?

Over the years, many different cell surface proteins and molecules have been reported to function as HCMV entry receptors or otherwise contribute to viral entry ([15,16,17,18,24,36,37,38,39,40,56,58,59,60,61,63,64,65,66,67,68,69,70], Table 1). Considering the plethora of different cell types that the virus infects, it seems plausible that many if not all of these molecules play bona fide roles in early events during infection. That said, certain cell surface molecules likely function as the primary entry receptors that physically interact with viral glycoprotein complexes to drive default modes of entry. These default modes, of course, appear to differ between target cell types. PDGFRα can be considered a primary entry receptor for trimer-dependent infection of fibroblasts, since wild-type virus is profoundly defective for entry into fibroblasts under conditions in which PDGFRα is absent or unavailable, and *pentamer*-null virus shows a more severe, virtually absolute entry defect in these settings [36,37,38,39]. Nrp2, on the other hand, can be considered a primary entry receptor for pentamer-dependent infection of epithelial and endothelial cells, since HCMV fails to infect these cell types when soluble Nrp2 is present, or when *Nrp2* is disrupted or knocked-down using small interfering RNA (siRNA) [40].

Receptors and co-receptors for viral entry often function at steps that are temporally and spatially distinct from each other [71]. For instance, a given receptor may interact with a viral glycoprotein complex at the cell surface to promote endocytosis of virions, while another cellular factor may be required for membrane fusion and escape from the endocytic compartment. Thus, although the evidence for co-receptors is less straightforward, cellular factors other than Nrp2 and PDGFRα may turn out to be absolutely required for downstream events during entry. Meanwhile, other cellular molecules may serve in secondary roles, wherein any one of a number of different proteins could substitute to promote entry. Finally, there may turn out to be examples in which a cellular factor promotes entry but is not strictly necessary for infection to occur.

Of course, the cell type being infected has implications for the mechanistic details at play. Trimer-dependent entry into fibroblasts is rapid, does not require clathrin, and is pH-independent, which led to the notion that this mode of entry involves fusion at plasma membrane [72,73] (Figure 1). According to the latest data, however, trimer-dependent entry into fibroblasts occurs through a rapid macropinocytosis [74]. Pentamer-dependent entry into epithelial and endothelial cells, on the other hand, requires low pH, and presumably involves a more prolonged form of endocytosis [73]. Overall, the literature has made clear that pentamer and trimer drive entry into different cell types via distinct cell surface receptors, which strongly suggests that are at least two major modes of HCMV entry: pentamer-dependent and trimer-dependent [36,37,40,75] (Figure 1). Each mode likely involves a unique set of host proteins that play cell type-specific roles as receptors, co-receptors, or accessory factors that enhance infection. Further complicating matters, it may turn out that pentamer-dependent entry relies on distinct co-receptors in epithelial cells versus endothelial cells, even if Nrp2 is a primary entry receptor in both settings.

## 9. How Do gH/gL Complexes Regulate Membrane Fusion?

It is assumed that upon recognition of the appropriate cell surface receptor, gH/gL complexes trigger gB to fuse virion and target cell membranes. Precisely how gH/gL complexes regulate the gB fusogen is nevertheless unclear. Data suggesting a physical interaction between gH/gL and gB come mainly from experiments with herpes simplex virus [76,77,78]. A recent HCMV study, however, reported co-immunoprecipitation (co-IP) results that suggest a stable complex between gB and gH/gL occurs in infected cells and in virions [79]. Excitingly, a very recent cryo-ET study of HCMV strain AD169 virions further bolsters the possibility that gH/gL and gB stably interact prior to fusion [14]. The study reports the *in situ* visualization of “L”-shaped densities that are interpreted to be gH/gL complexes in contact with pre-fusion gB trimers in the virion envelope [14]. Although such complexes could be modeled only at ~30 Å resolution, the authors were able to infer that domain I of gB may physically interact with gH. Roughly 7% of the putative gH/gL complexes appeared to contact pre-fusion gB homotrimers while no gH/gL complexes were observed to contact post-fusion gB. These findings are consistent with a model in which gH/gL stabilizes the pre-fusion conformation of gB until receptor binding events trigger the transition to the post-fusion state, which presumably drives membrane fusion events.

Since the trimer is strongly required for the infectivity of cell-free HCMV virions in all cell types, it has been argued that pentamer may stimulate endocytosis of virions, while actual membrane fusion events may require the trimer to activate gB [34]. Other authors argue that the trimer mediates interactions with cell surface molecules to promote virion uptake via macropinocytosis, while the pentamer is required for escape from endosomes [41]. In the latter scenario, the pentamer would presumably act as an alternative fusion trigger for gB. One might hypothesize that interactions with PDGFRα are required for the trimer to function as a fusion trigger for gB, and that in the absence of PDGFRα, the trimer promotes endocytosis but not membrane fusion. Future studies leveraging structural, biophysical, and cell biology-based approaches will be needed to illustrate how gB is regulated during pentamer/Nrp2-dependent versus trimer/PDGFRα-dependent modes of entry.

## 10. Cell-Associated versus Cell Free Spread

HCMV is thought to disseminate within the host primarily through direct cell-to-cell spread rather than via the release of extracellular (cell-free) virions, which would be susceptible to antibody responses. Most of the infectious virus in the blood of seropositive and acutely infected patients is found in the leukocyte compartment, rather than in plasma or serum [80,81,82]. Furthermore, clinical isolates of HCMV spread in a highly cell-associated manner during initial tissue culture passages [83,84], and the progressive loss of this cell-associated phenotype correlates with disruption of several elements within the viral genome [83,85]. Below, we consider some examples of viral genes that have been implicated to impact cell-associated versus cell-free spread.

### 10.1. RL13

*RL13*, which encodes a virion envelope glycoprotein, is among the viral genes that most rapidly mutate during tissue culture propagation of the virus, often acquiring nonsense or frameshift mutations after one to four passages on fibroblasts, endothelial cells, or epithelial cells [85,86]. Cultured fibroblasts infected with HCMV harboring repaired *RL13* and *UL128–UL131* loci produce remarkably small amounts of cell-free infectious virus until several weeks post-infection [85,86]. These observations may suggest that *RL13* functions to dampen HCMV spread in vivo, perhaps to promote long-term persistence. Alternatively, these findings may reflect selection pressures peculiar to laboratory cultivation of the virus under tissue culture conditions. Unfortunately, little is known about the function of *RL13*. Ectopically expressed RL13 has been shown to traffic to the cell surface and bind the F_C_ domain of IgG_1_ and IgG_2_ antibodies, followed by internalization [87]. These findings may imply an immune-evasive function for RL13, though RL13-dependent internalization of IgG has not yet been demonstrated in the context of the infected cell. RL13 has also been shown to strongly suppress the contribution of gO to cell-free spread in a *UL128*-null strain Merlin background [88]. Although these findings shed light on interactions of RL13 with cellular and viral factors, they do not readily explain why *RL13* is unstable during tissue culture propagation of HCMV.

### 10.2. gH/gL Complexes and Cell Tropism

The literature suggests that cell-free versus cell-associated modes of spread are governed in large part by the composition of gH/gL complexes expressed in the virion envelope. Restoration of pentamer expression increases the cell-associated nature of the virus [48], and a more recent study showed that the pentamer drives a mode of direct cell-to-cell spread that resists antibody neutralization [89]. Accordingly, repair of the pentamer in strain AD169 promotes the formation of syncytia during in vitro cultivation of the virus [42]. On the other hand, the trimer is required for the infectivity of cell-free virions [33,35]. Although one might assert that the pentamer promotes cell-associated modes of spread while the trimer enhances cell-free spread, the pentamer is nonetheless required for cell-free virions to efficiently infect endothelial and epithelial cells as well as monocytes. Another observation that would confound generalizing the trimer and pentamer into respective “cell-free” versus “cell-to-cell spread” roles is that HCMV strain AD169 deleted for the essential tegument protein pp28 (*UL99*) was found to spread efficiently in cultured fibroblasts [90]. Because strain AD169 harbors a frameshift in *UL131* that renders it unable to express pentamer, the efficient spread observed for the *pp28*-null virus has led some to argue that the trimer may suffice to drive cell-to-cell spread in fibroblasts [39].

### 10.3. Viral Genes and Polymorphisms that Impact HCMV Cell Tropism

HCMV strains show large differences in the relative levels of pentamer and trimer incorporated into virions, and these differences correlate with cell tropism differences between strains [34,91]. A number of HCMV genes have the capacity to influence viral cell tropism at the stage of entry, and most of these presumably act via effects on the composition of gH/gL complexes. Certain HCMV strains that maintain intact or largely intact UL*b’* regions express the pentamer at low levels. Examples include viruses derived from infectious bacterial artificial chromosome (BAC) clones of strains TR [92], TB40/E [93], and VR1814/FIX [94]. These strains, at least when reconstituted on fibroblasts, express high levels of gH/gL/gO (trimer), and low levels of gH/gL/UL128–131 (pentamer). Although a mutation in an intron of *UL128* has been identified to limit pentamer expression in strain TB40/E [95], why FIX and TR express similarly low levels of pentamer is unknown. Given their low levels of pentamer expression, it is perhaps unsurprising that these HCMV strains replicate inefficiently on epithelial cells [42]. On the other hand, the highly passaged strain AD169, which carries a UL*b’* region that has undergone rearrangements and deletions leading to loss of ~14 kbp of coding content [96], replicates robustly on epithelial cells when pentamer expression is restored [42].

Intriguingly, ablation of *UL148*, a gene within the UL*b’*, enhances the ability of strain TB40/E to replicate to high levels in ARPE-19 epithelial cells, and leads to a striking reduction in overall levels of gH/gL and of trimer in virions ([97,98]) (Figure 2). The *UL148*-null phenotype is accompanied by a markedly reduced expression of gO, which likely explains why *UL148*-null mutants express low levels of trimer. Nonetheless, the increase in epithelial cell tropism does not appear to involve enhanced pentamer expression.

We recently reported that newly synthesized gO, but not other glycoproteins, is constitutively targeted for ER-associated degradation (ERAD) during infection [98]. UL148 appears to reduce the rate at which gO is degraded, possibly by interacting with SEL1L, a core component of the ERAD machinery. The observation that gO behaves as a constitutive ERAD substrate suggests that modulation of ERAD could provide a platform for viral regulation of cell tropism in HCMV and perhaps other betaherpesviruses. Nonetheless, whether UL148 is somehow regulated to stabilize gO in a cell-type specific manner remains to be seen. Interestingly, UL148 was also recently identified to prevent surface presentation of CD58 (LFA-3), a co-stimulatory ligand for natural killer cells and T-cells [99], and to strongly contribute to the induction of the unfolded protein response (UPR) during infection [100]. It is not yet clear how the roles of UL148 in CD58 retention or gO stabilization relate to its activation of the UPR. However, pharmacologic and siRNA treatments that inhibit or deplete the ERAD machinery stabilize gO expression [98]. Since blockade of ERAD would also be expected to activate the UPR, it seems reasonable to hypothesize that UL148 functions in part by inhibiting ERAD.

US16 is another viral factor that was recently identified to impact the composition of gH/gL complexes in HCMV virions [101]. *US16*-null mutant viruses fail to incorporate pentamer into progeny virions and accordingly are unable to efficiently infect epithelial cells or endothelial cells. Unlike UL148, which resides in the ER, US16 localizes to the cytoplasmic viral assembly compartment (cVAC), where virions acquire their infectious envelopes. How US16 promotes pentamer incorporation is unclear, but co-immunoprecipitation results suggest that US16 interacts with the pentamer subunit UL130. The observation that US16 localizes to—and presumably functions at—the cVAC may suggest that the trans-Golgi network-derived vesicles which provide virion envelopes are heterogeneous. Alternatively, US16 may negatively regulate degradation of pentamer by cVAC-associated lysosomes, or otherwise promote incorporation of the pentamer into virions.

It is intriguing to speculate that gene products such as US16 and UL148 could play tissue-specific roles in modulating the composition of viral gH/gL complexes in vivo. The relative expression levels of pentamer and trimer may influence whether HCMV infection spreads in a cell-associated, antibody-resistant manner, or via cell-free particles, as would be more compatible with horizontal transmission via body fluids. It thus seems crucial to identify how viral modulators of the alternative gH/gL complexes might themselves be regulated. Of course, it would be helpful to know in the first place whether virions shed in saliva, urine, or breast milk contain higher levels of trimer than those produced for intra-host spread.

In the example of EBV, a viral tropism switch that drives alternating cycles of viral replication between B-cells and epithelial cells depends on the cell type producing virus to regulate the levels in of gH/gL and gH/gL/gp42 in progeny virions, with the gp42 containing complex being required for infection of B-cells [102]. Although the cell tropism of fibroblast-derived HCMV virions is reportedly more heterogeneous than that of virions produced from endothelial cells [2], additional research in this area is certainly warranted. Regardless, should a tropism switch exist in HCMV, one might expect the mechanisms to differ substantially from those found in EBV.

## 11. What Can We Infer from Cytomegaloviruses of Rodents and Nonhuman Primates?

A large number of CMVs co-speciated with their hosts during the mammalian radiation, which occurred approximately 60 to 80 million ago [103]. Accordingly, rodent and primate CMVs are studied in the context of their natural hosts to model myriad aspects of HCMV biology and pathogenesis [104,105,106,107,108,109], as well as to evaluate the potential use of HCMV as a vaccine vector [110,111]. Even though primate and rodent CMVs share many features, including their tropism for the salivary gland [112,113], the extent to which viral entry and host navigation are conserved remains largely unknown. Although CMVs are thought to enter new hosts either via the oral cavity (ingestion) or via the nasal route (inhalation), the recent literature suggests that MCMV strongly relies on the nasal route to establish infection [114]. Following intranasal infection of mouse pups, a *gO*-null MCMV mutant was unable to colonize the salivary gland, suggesting that systemic spread depends on gO [115]. Furthermore, these defects were associated with an impaired ability of the *gO*-null virus to infect epithelial cells [115]. In contrast, a *gO*-null virus was observed to readily colonize the salivary gland in immunocompromised BALB/C mice infected intraperitoneally [116]. Therefore, the route of infection may have strong effects on tissue tropism phenotypes.

Given that millions of years have elapsed since the ancestral cytomegalovirus infected the common ancestor of tree shrews, rodents and primates [103,117], it is not surprising to find examples in which the extant CMVs have solved similar problems related to host colonization in different ways. For example, although HCMV and rhesus CMV (RhCMV) both encode ER-resident immuneëvasins that block cell surface presentation of natural killer cell activating ligands of the MIC and ULBP families, this function is carried out by UL16 in HCMV [118,119,120] but by Rh159 (the homolog of UL148) in RhCMV [121]. Nevertheless, the locus encoding gO (*UL74*) and gN (*UL73*), in which the residues encoding the C-termini of the encoded proteins overlap, is remarkably well-conserved across rodent and primate cytomegaloviruses [86,92,122,123,124,125,126], as is the use of alternative gH/gL complexes. On the other hand, the HCMV UL*b’* region [96], which encodes pentamer components UL128-131, the putative tropism modulator UL148, and several other proteins with roles in persistence, differs substantially in rhesus CMV [127] and is virtually unrecognizable in rodent CMVs [122,123,124]. In mouse cytomegalovirus (MCMV), the alternative gH/gL complexes identified thus far are both trimeric. One is comprised of gH/gL bound to gO (m74) [128]; the other is made up of gH/gL bound to MCK-2, a chemokine homolog [129]. RhCMV and guinea pig CMV (GPCMV), however, appear to make use of pentameric and trimeric gH/gL complexes, and hence more closely resemble HCMV in this regard [127,130,131,132,133,134]. The two rat CMVs, which in fact represent distinct species [135], remain to be characterized for whether a pentameric complex or one akin to MCMV gH/gL/MCK-2 is expressed in virions [136].

Needless to say, it will be crucial to determine whether the animal orthologs of Nrp2 and PDGFRα function as entry receptors for the alternative gH/gL complexes in CMVs of rodents and nonhuman primates. Should the underlying details for viral entry prove to be broadly conserved across the rodent and primate CMVs, this would further buttress the clinical relevance of the animal CMVs infection models.

## 12. Conclusions and Outlook

Although HCMV entry and host navigation remain to be fully understood, crucial new details concerning cellular receptors and the viral entry machinery have no doubt invigorated the field. It is hoped that the coming decade will see investigators leverage these advances to elucidate how gB-mediated membrane fusion events are regulated in response to interactions between viral glycoprotein complexes and their cognate cellular receptors, as well as to reveal how HCMV spreads throughout the host to reach sites relevant to persistence and horizontal shedding. Such information will be pivotal for the development of vaccines and therapies to combat the diseases caused by this fascinating opportunistic pathogen.

## Figures and Tables

**Figure 1 viruses-10-00704-f001:**
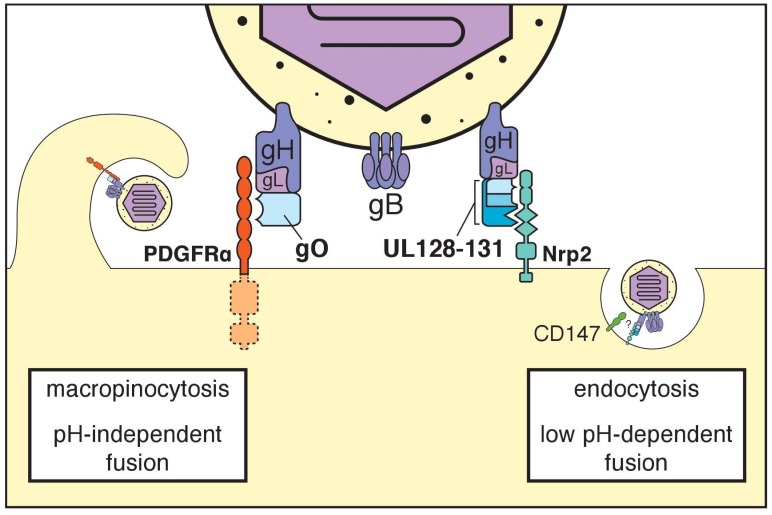
Receptors for HCMV gH/gL complexes. The trimeric gH/gL/gO complex (trimer) interacts with PDGFRα to drive a pH-independent mode of entry that involves macropinocytosis. The pentameric gH/gL/UL128–131 complex (pentamer) interacts with Nrp2 in a mode of entry that involves endocytosis and a decrease in pH. CD147 also appears to be required in the latter mode of entry. See text for additional details.

**Figure 2 viruses-10-00704-f002:**
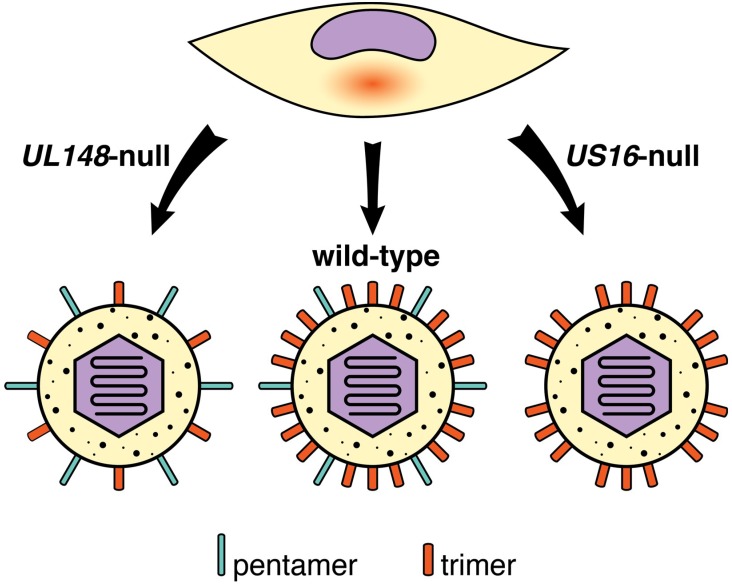
Regulation of alternative gH/gL complexes by UL148 and US16. UL148, a viral endoplasmic reticulum (ER)-resident glycoprotein, promotes high-level expression of the trimer during infection (wild-type) by stabilizing gO within the endoplasmic reticulum, resulting in the production of trimer-rich progeny virions. In *UL148*-null infections, decreased levels of trimer are synthesized, leading to the production of virions that more efficiently infect and spread between epithelial cells. US16, in contrast, localizes to the viral assembly compartment to promote incorporation of the pentamer into progeny virions, perhaps via physical interaction with UL130. *US16*-null mutants produce progeny virions lacking pentamer that are unable to infect endothelial and epithelial cells. See text for additional details.

**Table 1 viruses-10-00704-t001:** Host cell surface factors implicated in human cytomegalovirus (HCMV) entry.

Host Cell Surface Entity	References
Heparan sulfate proteoglycans (HSPG)	[17,24,63,64]
Platelet-derived growth factor receptor alpha (PDGFRα)	[16,36,37,38,39,65]
Neuropilin-2 (Nrp2)	[40]
Epidermal growth factor receptor (EGFR)	[15,66,67,68]
αVβ3 integrin	[18,61,68]
α2β1 integrin; α6β1 integrin	[17,18,68]
α1β1 integrin; α3β1 integrin	[68]
Major histocompatibility complex class I (MHC-I)	[69]
CD13 (alanyl aminopeptidase)	[70]
CD90 (THY-1)	[59,60]
CD147 (Basigin)	[56]
CD151 (MER2, RAPH, PETA-3)	[58]

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
