# Peer review of "Pathogen at the Gates: Human Cytomegalovirus Entry and Cell Tropism"

_viruses, 2018, doi:10.3390/v10120704_

Round 1

Reviewer 1 Report

In this manuscript, Nguyen and Kamil describe the state of the art of the human cytomegalovirus entry pathways, covering a subject that has recently received new inputs and that promises new developments in a closer future. The review is well written, the literature is up-to-date, and a valuable historical reconstruction of the major milestones has been satisfactorily achieved. While I have no comments against publication of the article, and recommend acceptance without modification, I do have a general comment for the authors. 

1.    In my opinion, the section on “cell-associated versus cell free spread” (lines 230-270) is a weak part of the review and could have been approached more extensively. In fact, in the last years the list of HCMV envelope proteins has grown, although most of the newly identified species remain to be functionally characterized. For example, the authors pay due attention to the role of RL13, strongly supported by genetic evidences, but the hypothetical involvement of at least two other HCMV Fc binding proteins (UL119 and RL11) could have been discussed in relation to what is known for other herpesiviridae. In HSV-1, gE-gI are part of the virion and function not only as Fc receptor but, at least gE, has a crucial role in cell-to-cell spread. Same situation occurs in VZV, where the importance of the gE subunit is amplified by the fact that it alone represents the commercial vaccine antigen. I am certain the authors are aware of these aspects and suppose that space limitations imposed focusing on the “major viral genes known to impact cell-associated vs cell-free spread”. The foregoing could be a suggestion for a future review.

Author Response

Reviewer 1

In this manuscript, Nguyen and Kamil describe the state of the art of the human cytomegalovirus entry pathways, covering a subject that has recently received new inputs and that promises new developments in a closer future. The review is well written, the literature is up-to-date, and a valuable historical reconstruction of the major milestones has been satisfactorily achieved. While I have no comments against publication of the article, and recommend acceptance without modification, I do have a general comment for the authors. 

1.    In my opinion, the section on “cell-associated versus cell free spread” (lines 230-270) is a weak part of the review and could have been approached more extensively. In fact, in the last years the list of HCMV envelope proteins has grown, although most of the newly identified species remain to be functionally characterized. For example, the authors pay due attention to the role of RL13, strongly supported by genetic evidences, but the hypothetical involvement of at least two other HCMV Fc binding proteins (UL119 and RL11) could have been discussed in relation to what is known for other herpesiviridae. In HSV-1, gE-gI are part of the virion and function not only as Fc receptor but, at least gE, has a crucial role in cell-to-cell spread. Same situation occurs in VZV, where the importance of the gE subunit is amplified by the fact that it alone represents the commercial vaccine antigen. I am certain the authors are aware of these aspects and suppose that space limitations imposed focusing on the “major viral genes known to impact cell-associated vs cell-free spread”. The foregoing could be a suggestion for a future review.

We agree that this section could be expanded.  However, because as the reviewer acknowldeges “most of the newly identified species remain to be functionally characterized” we thought including such informaiton might be a bit too speculative.  The analogy to VZV and HSV is also exicting and fascinating, but because of the huge evolutionary distance between the alpha- and beta- herpesviruses, we feel it might not make a for a valid comparison.  We do apprecaite the suggestion that this would make an excellent topic for a future review.  

We are very grateful to this reviewer as well as to reviewers 2 and 3 for taking the time to read our manuscript and to provide critical feedback. 

Reviewer 2 Report

In this manuscript, the authors provide substantial information on HCMV cell tropism in relevance to viral entry machinery, with a particular focus on the relations between the viral envelope complexes and their cellular receptors. The review is overall clear and well-written, easy to read and understand, and the authors summarize current data in accordance with the recent literature. The references are properly formatted. The catching title deserves an attention.

Minor points:

- Line 157: a comma is missing;

- Line 196-198: please check the sentence;

- Line 121: what about epithelial cells, such as keratinocytes or Arpe-19? Is UL128-131 locus dispensable?

- Line 208-212: please clarify the differences between endothelial and epithelial cells and the relative role of Nrp2;

- Conclusion and Outlook: antiviral drug research targeting HCMV entry should be discussed or at least cited;

- finally, a summarizing table on the main complexes and receptors, with a small description AKA bullet points, will help the reader.

Author Response

Reviewer 2

In this manuscript, the authors provide substantial information on HCMV cell tropism in relevance to viral entry machinery, with a particular focus on the relations between the viral envelope complexes and their cellular receptors. The review is overall clear and well-written, easy to read and understand, and the authors summarize current data in accordance with the recent literature. The references are properly formatted. The catching title deserves an attention.

Minor points:

- Line 157: a comma is missing;

this has been corrected, we thank the reviewer for alerting us of this typo. 

- Line 196-198: please check the sentence;

this sentence has also been revised. thanks for catching the issue.

- Line 121: what about epithelial cells, such as keratinocytes or Arpe-19? Is UL128-131 locus dispensable?

these points have been clarified — to make more clear that UL128-131 is not dispensible for entry to epithelial cells but is dispensible for infection of fibroblasts. 

- Line 208-212: please clarify the differences between endothelial and epithelial cells and the relative role of Nrp2;’

To our knowledge Martinez-Martin et al found evidence that Nrp2 is required for entry into both cell types, and we thought this assumption was reflected in our writing here.  Could the reviewer please let us know what differences he or she is referring to? 

- Conclusion and Outlook: antiviral drug research targeting HCMV entry should be discussed or at least cited;

we are unable to find much in the literature on drugs targeting viral entry, other than a report by Teresa Compton’s group (English et al, 2006. Journal of Biological Chemistry, 281(5), pp.2661-2667.)   It would seem out-of-place to cite this one study on its own, and we do not know of any ongoing pre-clinical development of the compounds studied in that paper. 

- finally, a summarizing table on the main complexes and receptors, with a small description AKA bullet points, will help the reader.

We agree that this would be helpful.  Please see the inclusion now of Table 1 (and supporting references), as per the reviewer’s request.    Note, however, that we felt it too contentious and somewhat impractical for other reasons to include the viral glycoprotein complexes.   Experimental evidence matching a complex with its receptor is not always provided in the literature, and it would be difficult to organize a table where that information was left out in certain places but included in others .. plus it would really highlight where certain authors were incorrect and would accentuate ongoing unresolved controveries in the field (hence why it would be contentious) ,

Reviewer 3 Report

My only suggestion is that the work does not mention CMV of other species...notably mouse which has been studied extensively. As the pattern of cellular tropism is similar, information can be gleaned from genetic similarities and differences between HCMV and MCMV. This could be addressed in an additional  paragraph.

I also wonder about the choice of "other candidates". I am not clear why Thy-1 is highlighted in this context since many Thy-1+ cells are never infected by CMV. The section omits many other candidates such as MHC class I  [Immunol Today. 1994;15(6):295-6], heparan sulphate etc.

Author Response

Reviewer 3

My only suggestion is that the work does not mention CMV of other species...notably mouse which has been studied extensively. As the pattern of cellular tropism is similar, information can be gleaned from genetic similarities and differences between HCMV and MCMV. This could be addressed in an additional  paragraph.

I also wonder about the choice of "other candidates". I am not clear why Thy-1 is highlighted in this context since many Thy-1+ cells are never infected by CMV. The section omits many other candidates such as MHC class I  [Immunol Today. 1994;15(6):295-6], heparan sulphate etc. 

We thank this reviewer for their helpful comments and suggestion. 

We have added a section about CMVs of other species, see lines 362-401 of the revision.    We hope the inclusion of Table 1 and inclusion of additional references now cited on 187-188 (in support of MHC-I and other molecules as possible HCMV receptors) will also please this reviewer.    Otherwise, the reason THY-1 was highlighted is because two recent reports from Jeffrey Cohen’s lab at NIAID report it as a receptor and the major impetus  for this review article was to summarize recent advances in the area of HCMV tropism.